# TIME LIMITS IN REINFORCEMENT LEARNING

## ABSTRACT

In reinforcement learning, it is common to let an agent interact with its environment for a fixed amount of time before resetting the environment and repeating the process in a series of episodes. The task that the agent has to learn can either be to maximize its performance over (i) that fixed amount of time, or (ii) an indefinite period where the time limit is only used during training. In this paper, we investigate theoretically how time limits could effectively be handled in each of the two cases. In the first one, we argue that the terminations due to time limits are in fact part of the environment, and propose to include a notion of the remaining time as part of the agent's input. In the second case, the time limits are not part of the environment and are only used to facilitate learning. We argue that such terminations should not be treated as environmental ones and propose a method, specific to value-based algorithms, that incorporates this insight by continuing to bootstrap at the end of each partial episode. To illustrate the significance of our proposals, we perform several experiments on a range of environments from simple few-state transition graphs to complex control tasks, including novel and standard benchmark domains. Our results show that the proposed methods improve the performance and stability of existing reinforcement learning algorithms.

## 1 INTRODUCTION

The reinforcement learning framework (Sutton & Barto, 1998; Bertsekas & Tsitsiklis, 1996; Szepesvari, 2010; Kaelbling et al., 1996) involves a sequential interaction between an agent and its environment. At every time step $t$, the agent receives a representation $S_t$ of the environment's state, selects an action $A_t$ that is executed in the environment which in turn provides a representation $S_{t+1}$ of the successor state and a reward signal $R_{t+1}$. An individual reward received by the agent does not directly indicate the quality of its latest action as some rewards may indeed be the consequence of a series of actions taken far in advance. Thus, the goal of the agent is to learn a good policy by maximizing the discounted sum of future rewards also known as *return*:

$$G_t^{\gamma} \doteq R_{t+1} + \gamma R_{t+2} + \gamma^2 R_{t+3} + ... = \sum_{k=0}^{\infty} \gamma^k R_{t+k+1} = R_{t+1} + \gamma G_{t+1}^{\gamma} \tag{1}$$

A discount factor $0 \le \gamma < 1$ is necessary to exponentially decay the future rewards ensuring bounded returns. While the series is infinite, it is common to use this expression even in the case of possible terminations. Indeed, episode terminations can be considered to be the entering of an absorbing state that transitions only to itself and generates zero rewards thereafter. However, when the maximum length of an episode is predefined, it is easier to rewrite the expression above by explicitly including the time limit $T$:

$$G_{t:T}^{\gamma} \doteq R_{t+1} + \gamma R_{t+2} + \gamma^2 R_{t+3} + ... + \gamma^{T-t-1} R_T = \sum_{k=0}^{T-t-1} \gamma^k R_{t+k+1} = R_{t+1} + \gamma G_{t+1:T}^{\gamma} \tag{2}$$

Optimizing for the expectation of the return specified in Equation 2 is suitable for naturally *time-limited tasks* where the agent has to maximize its expected return $G_{0:T}$ over a fixed episode length only. In this case, since the return is bounded, a discount factor of $\gamma = 1$ can be used. However, in practice it is still common to keep $\gamma$ smaller than 1 in order to give more priority to short-term rewards. Under this optimality model, the objective of the agent does not go beyond the time limit. Therefore, an agent optimizing under this model should ideally learn to take more risky actions that

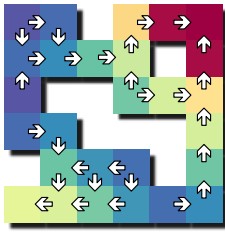 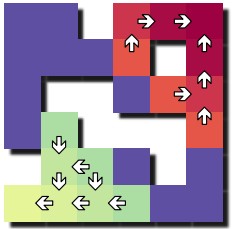 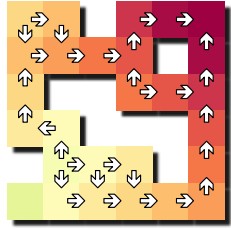

(a) Standard        (b) Time-awareness       (c) Partial-episode bootstrapping

Figure 1: Illustrations of color-coded state-values and policies overlaid on our Two-Goal Gridworld problem with two rewarding terminal states (50 for reaching the top-right corner and 20 for the bottom-left), a penalty of $-1$ for moving, and a time limit $T = 3$. (a) A standard agent without time-awareness which cannot distinguish between timeout terminations and environmental ones. (b) An agent with the proposed time-awareness that learns to stay in place when there is not enough time to reach a goal. (c) An agent with the proposed partial-episode bootstrapping that continues to bootstrap from any early terminations in order to maximize its return over an indefinite period.

reveal higher expected return than safer ones as approaching the end of the time limit. In Section 2, we study this case and illustrate that due to the presence of the time limit, the remaining time is present in the environment's state and is essential to its *Markov property* (Sutton & Barto, 1998). Therefore, we propose to include a notion of the remaining time in the agent's input, an approach that we refer to as *time-awareness*. We describe various general scenarios where lacking a notion of the remaining time can lead to suboptimal policies and instability, and demonstrate significant performance improvements for time-aware agents.

Optimizing for the expectation of the return specified by Equation 1 is relevant for *time-unlimited tasks* where the interaction is not limited in time by nature. In this case, the agent has to maximize its expected return over an indefinite (e.g. infinite) period. However, it is desirable to use time limits in order to diversify the agent's experience. For example, starting from highly diverse states can avoid converging to suboptimal policies that are limited to a fraction of the state space. In Section 3, we show that in order to learn good policies that continue beyond the time limit, it is important to differentiate between the terminations that are due to time limits and those from the environment. Specifically, for value-based algorithms, we propose to continue bootstrapping at states where termination is due to the time limit, or generally any other causes other than the environmental ones. We refer to this method as *partial-episode bootstrapping*. We describe various scenarios where having a time limit can facilitate learning, but where the aim is to learn optimal policies for indefinite periods, and demonstrate that our method can significantly improve performance.

We evaluate the impact of the proposed methods on a range of novel and popular benchmark domains using a deep reinforcement learning (Arulkumaran et al., 2017; Henderson et al., 2017) algorithm called the Proximal Policy Optimization (PPO), one which has recently been used to achieve state-of-the-art performance in many domains (Schulman et al., 2017; Heess et al., 2017). We use the OpenAI Baselines[1] implementation of the PPO algorithm with the hyperparameters reported by Schulman et al. (2017), unless explicitly specified. All novel environments are implemented using the OpenAI Gym framework (Brockman et al., 2016) and the standard benchmark domains are from the MuJoCo (Todorov et al., 2012) Gym collection. We modified the TimeLimit wrapper to include remaining time in the observations for the proposed time-aware agent and a flag to separate timeout terminations from environmental ones for the proposed partial-episode bootstrapping agent. For every task involving PPO, to have perfect reproducibility, we used the same 40 seeds from 0 to 39 to initialize the pseudo-random number generators for the agents and environments. Every 5 training cycles (i.e. 10240 time steps), we perform an evaluation on a complete episode and store the sums of rewards, discounted returns, and estimated state-values. For generating the performance plots, we average the values across all runs and then apply smoothing with a sliding window of size 10. The performance graphs show these smoothed averages as well as their standard error.

We empirically show that time-awareness significantly improves the performance and stability of PPO for the time-limited tasks and can sometimes result in quite interesting behaviors. For example,

---

[1] https://github.com/openai/baselines

in the Hopper-v1 domain with $T = 300$, our agent learns to efficiently jump forward and fall towards the end of its time in order to maximize its travelled distance and achieve a "photo finish". For the time-unlimited tasks, we show that bootstrapping at the end of partial episodes allows to significantly outperform the standard PPO. In particular, on Hopper-v1, even if trained with episodes of only 200 steps, our agent manages to learn to hop for at least $10^6$ time steps, resulting in more than two hours of rendered video. Detailed results for all variants of the tasks using PPO with and without the proposed methods are available in the Appendix. The source code will be made publicly available shortly. A visual depiction of highlights of the learned behaviors can be viewed at the address sites.google.com/view/time-limits-in-rl.

## 2 TIME-AWARENESS FOR TIME-LIMITED TASKS

In tasks that are time-limited by nature, the learning objective is to optimize the expectation of the return $G_{0:T}^{\gamma}$ from Equation 2. Interactions are systematically terminated at a fixed predetermined time step $T$ if no environmental termination occurs earlier. This time-wise termination can be seen as transitioning to a terminal state whenever the time limit is reached. The states of the agent's environment, formally a *Markov decision process* (MDP) (Puterman, 2014), thus must contain a notion of the remaining time that is used by its transition function. This time-dependent MDP can be thought of as a stack of $T$ time-independent MDPs, one for each time step, followed by one that only transitions to a terminal state. Thus, a decision-making agent in such an environment, at every time step $t \in \{0, ..., T-1\}$, takes a decision that results in transitioning to a new state from the next MDP in the stack and receiving a reward.

Thus, a time-unaware agent effectively has to act in a *partially observable Markov decision process* (POMDP) (Lovejoy, 1991) where states that only differ by their remaining time appear identical. This phenomenon is a form of *state-aliasing* (Whitehead & Ballard, 1991) that is known to lead to suboptimal policies and instability due to the infeasibility of correct *credit assignment*. In this case, the terminations due to time limits can only be interpreted as part of the environment's stochasticity where the time-unaware agent perceives a chance of transitioning to a terminal state from any given state. In fact, this perceived stochasticity is dynamically changing with the agent's behavioral policy. For example, an agent could choose to stay in a fixed initial state during the entire course of an episode and perceive the probability of termination from that state to be $\frac{1}{T}$, whereas it could choose to always move away from it in which case this probability would be perceived to be zero.

In the view of the above, we propose time-awareness for reinforcement learning agents in time-limited domains by including directly the remaining time $T - t$ in the agent's representation of the environment's state or by providing a mean to infer it. The importance of the inclusion of a notion of time in time-limited problems was first demonstrated by Harada (1997), but seems to have been overlooked in the design of the benchmark domains and the evaluation of reinforcement learning agents. A major difference between the approach of Harada (1997) and that of ours, however, is that we consider a more general class of time-dependent MDPs where the reward distribution and the transitions can also be time-dependent, preventing the possibility to consider multiple time instances at once as it is the case for the $Q_T$-learning algorithm (Harada, 1997).

Here, we illustrate the issues faced by time-unaware agents by exemplifying the case for value-based methods. The state-value function for a time-aware agent in an environment with time limit $T$ is:

$$v_\pi(s, \boldsymbol{T - t}) \doteq \mathbb{E}_\pi \left[ G_{t:T}^\gamma | S_t = s \right] \tag{3}$$

By denoting an estimate of the state-value function by $\hat{v}_\pi$, the *temporal-difference* (TD) update rule (Sutton, 1988) after transitioning from a state $s$ to a state $s'$ and receiving a reward $r$ as a result of an action $a$ is given by either of the following expressions conditioned on the time step $t$:

$$
\begin{aligned}
\hat{v}_\pi(s, \boldsymbol{T - t}) &\leftarrow (1-\alpha)\hat{v}_\pi(s, \boldsymbol{T - t}) + \alpha \left( r + \gamma \hat{v}_\pi(s', \boldsymbol{T - t + 1}) \right) && \text{for } t < T-1 \\
\hat{v}_\pi(s, \boldsymbol{1}) &\leftarrow (1-\alpha)\hat{v}_\pi(s, \boldsymbol{1}) + \alpha r && \text{for } t = T-1
\end{aligned} \tag{4}
$$

The proposed added notion of the remaining time is indicated in bold and blue. A time-unaware agent would be deprived of this information and thus would update $\hat{v}_\pi(s)$ with or without bootstrapping from the estimated value of $s'$ depending on whether the time limit is reached. Confused by the conflicting updates for estimating the value of the same state, instead of learning an accurate value function, this time-unaware agent learns an approximate average of these inconsistent updates.

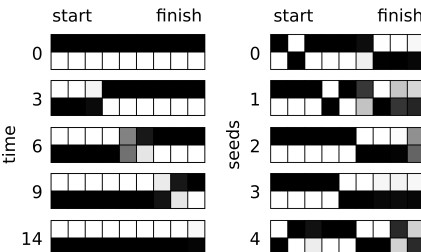

Figure 2: The color-coded learned action probabilities overlaid on our Queue of Cars problem (black and white indicate 0 and 1, respectively). For each block, the top row represents the dangerous action and the bottom row the safe one. The 9 non-terminal states are represented horizontally. Left: a time-aware PPO agent at various times: the agent learns to optimally select the dangerous action. Right: 5 different instances of the time-unaware PPO agent.

If the time limit $T$ is never varied, inclusion of the time $t$ as a notion of the remaining time would be sufficient. However, for more generality we choose to represent the remaining time $T-t$. In practice, we used the remaining time normalized from 1 to 0, concatenated to the observations provided by the Gym environments by modifying the TimeLimit wrapper.

## 2.1 THE LAST MOMENT PROBLEM

To give a simple example of the learning of an optimal time-dependent policy, we consider an MDP containing two states A and B. The agent always starts in A and has the possibility to choose an action to "stay" in place with no rewards or a "jump" action that transitions it to state B with a reward of $+1$. However,

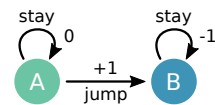

state B is considered a trap where the only possible action leads to a penalty of $-1$. The episodes terminate after a fixed number of steps $T$. The goal of the game is thus to jump at the last moment. For a time-unaware agent, the task is impossible to master for $T > 1$ and the best feasible policy would be to stay in place, resulting in an overall return of 0. In contrast, a time-aware agent can learn to stay in place for $T - 1$ steps and then jump, scoring an undiscounted sum of rewards of $+1$.

## 2.2 THE TWO-GOAL GRIDWORLD PROBLEM

To further illustrate the impact of state-aliasing for time-unaware agents, we consider a deterministic gridworld environment (see Figure 1) consisting of two possible goals rewarding 50 for reaching the top-right and 20 for the bottom-left. The agent has 5 actions: to move in cardinal directions or to stay in place. Any movement incurs a penalty of $-1$ while staying in place generates a reward of 0. Episodes terminate via a timer at $T = 3$ or if the agent has reached a goal. The initial state is randomly selected for every episode, excluding goal states. For training, we used a tabular Q-learning (Watkins &

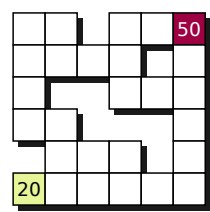

Dayan, 1992) with completely random actions, trained until convergence with a decaying learning rate and a discount factor of $\gamma = 0.99$.

The time-aware agent has a state-action value table for each time step and easily learns the optimal policy which is to go for the closest goal when there is enough time, and to stay in place otherwise. For the time-unaware agent, the greedy values of the cells adjacent to the goal with the terminal reward of 50 converge to 49 and those adjacent to the goal with 20 converge to 19 because of the $-1$ penalty on every move. Then, since the time limit is $T = 3$, from each remaining cell, the agent may have between 1 and 3 steps. For $2/3$ of the times, the time-unaware agent receives $-1$ and bootstraps from the successor cell and for $1/3$ of the times it receives $-1$ and experiences termination. Thus, for $v(s) = \arg\max_a q(s, a)$ and $N(s)$ denoting the neighbors of $s$, for states non adjacent to the goals we have: $v(s) = 2/3(-1 + \gamma \max_{s' \in N(s)} v(s')) + 1/3(-1)$. This learned value function leads to a policy that always tries to go for the closest goal even if there is not enough time. While the final optimal policy does not require time information, this example clearly shows that the confusion during training due to state-aliasing can create a leakage of the values to states that are out of reach.

It is worth noting that, Monte Carlo methods such as REINFORCE (Williams, 1992; Sutton et al., 2000) are not susceptible to this leakage as they use complete returns instead of bootstrapping. However, without awareness of the remaining time, Monte Carlo methods would still not be able to learn an optimal policy in many cases, such as the Last Moment problem or the Queue of Cars problem in the subsequent section.

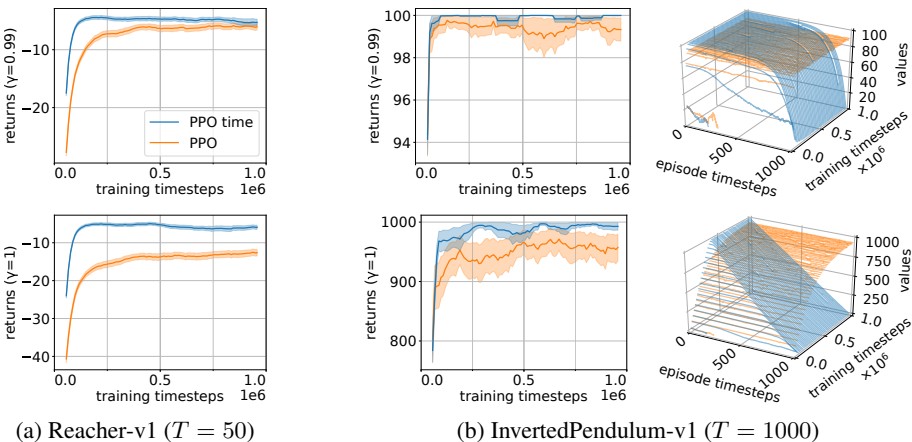

(a) Reacher-v1 ($T = 50$)                    (b) InvertedPendulum-v1 ($T = 1000$)

Figure 3: Comparison of PPO with and without the remaining time in input. (a) Performance on the Reacher-v1. (b) Performance on the InvertedPendulum-v1 (Left) and the learned state-value estimations against episodic time steps and training progress (Right). Top: The results for $\gamma = 0.99$. Bottom: The result for $\gamma = 1$.

## 2.3  THE QUEUE OF CARS PROBLEM

An interesting property of time-aware agents is the ability to dynamically adapt to the remaining time that can, for example, be correlated with the current progress of the agent. To illustrate this, we introduce an environment which we call Queue of Cars where the agent controls a vehicle that is held up be-

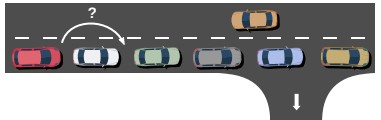

hind an intermittently moving queue of cars. The agent's goal is to reach an exit located 9 slots away from its starting position. At any time, the agent can choose the "safe" action to stay in the queue which may result in advancing to the next slot with 50% probability. Alternatively, it has the possibility to attempt to overtake by a "dangerous" action that even though it has 80% probability to advance, it poses a 10% chance of collision with the oncoming traffic and terminating the episode. The agent receives no rewards unless it reaches its destination to receive a +1 reward. The episode terminates by reaching the destination, running out of time, or colliding during an overtake.

In this task, an agent can have a lucky sequence of safe transitions and reach the destination within the time limit without ever needing to attempt an overtake. However, the opposite can also happen in which case the agent would need to overtake the cars to reach its destination in time. Time-unaware agents cannot possibly gauge the necessity to rush and thus can only learn a statistically efficient combination of dangerous and safe actions based on position only. Figure 2 shows this situation for a time-unaware PPO over 5 different runs against a time-aware one that adapts to the remaining time based on its distance to the goal to take more dangerous actions in the face of time insufficiency. A discount factor of $\gamma = 1$ was used for both agents.

## 2.4  STANDARD CONTROL TASKS

In this section, we evaluate the performance of PPO with and without the remaining time as part of the agent's input on a set of deterministic, continuous control tasks from the OpenAI's MuJoCo Gym benchmarks (Brockman et al., 2016; Todorov et al., 2012; Duan et al., 2016). By default, these environments

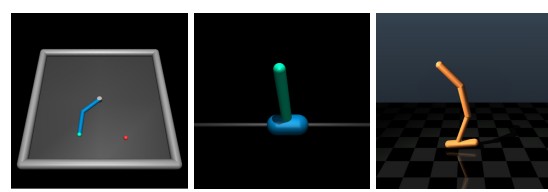

use predefined time limits and are each reset to a random initial state after an episode termination.

Figure 3 shows the performance of a time-unaware PPO against a time-aware one, demonstrating that time-awareness significantly improves the performance of PPO. The learned state-values shown for the InvertedPendulum-v1 task (see Figure 3b) illustrate perfectly the difference between a time-

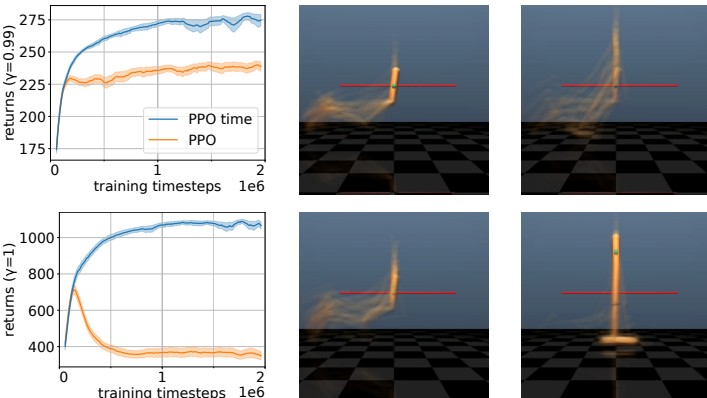

Figure 4: Comparison of PPO with and without the remaining time in input on Hopper-v1 ($T = 300$). Left: Performance evaluations. Middle: The average last pose of the time-aware PPO agent, reproduced with aligned $x$-axes. A green mark indicates the last measured $y$ coordinate of the agent used in the task with the termination threshold of $0.7$ meters indicated with a red line. Right: The average last pose of the time-unaware PPO. Top: Discount factor of $0.99$. Bottom: Discount factor of $1$. The different instances of the time-aware agent learn to jump forward before the time limit. A larger discount factor highly destabilizes the time-unaware PPO.

aware agent and a time-unaware one in terms of their estimated expected return as the episode progresses. While time-awareness enables the agent to learn an accurate exponential or linear decay of the expected return with time, the time-unaware agent only learns a constant estimate due to state-aliasing.

Figure 4 (left) shows the performance comparisons of PPO with and without time-awareness in the Hopper-v1 domain with time limit $T = 300$. With a discount rate of $0.99$, the standard PPO is initially on par with the time-aware PPO and later starts to plateau. As the agents become better, they start to experience terminations due to the time limit more frequently, at which point the time-unaware agent begins to perceive inconsistent returns for seemingly similar states. The advantage of the time-aware PPO becomes even clearer in the case of a discount rate of $1$ where the time-unaware PPO diverges quite drastically. A possible reason is that the time-unaware PPO agent experiences much more significant conflicts as the returns are now the sum of the undiscounted rewards. This is while, the time-aware PPO still continues to perform well as it is able to assign credits appropriately based on the knowledge of the remaining time.

Time-awareness does not only help agents by avoiding the conflicting updates. In fact, in naturally time-limited tasks where the agents have to maximize their performance for a limited time, time-aware agents can demonstrate quite interesting ways of ensuring to achieve this objective. Figure 4 show the average final pose of the time-aware (middle) and time-unaware (right) agents. We can see that the time-aware agent learns to jump towards the end of its time in order to maximize its expected return, resulting in a "photo finish", something that a time-unaware agent cannot accurately achieve. Finally, Figure 4 (bottom-right) shows an interesting behavior robustly demonstrated by the time-unaware PPO in the case of $\gamma = 1$ that is to actively stay in place, accumulating at least the rewards coming from the bonus for staying alive.

In this section, we explored the scenario where the aim is to learn a policy that maximizes the expected return over a limited time. We proposed to include a notion of the remaining time as part of the agent's observation to avoid state-aliasing which can cause suboptimal policies and instability. However, this scenario is not always ideal as there are cases where, even though the agent experiences time limits in its interaction with the environment, the objective is to learn a policy for a time-unlimited task. For instance, as we saw in the Hopper environment, the learned policy that maximizes the return over the $T = 300$ time steps generally results in a photo finish which would lead to a fall and subsequent termination if the simulation was to be extended. Such a policy is not viable if the goal is to learn to move forward for an indefinite period of time. One solution is to not have time limits during training. However, it is often more efficient to instead have short snippets of

interactions to expose the agent to diverse experiences. In the next section, we explore this case and propose a method that enables to effectively learn in such domains from partial episodes.

## 3 PARTIAL-EPISODE BOOTSTRAPPING FOR TIME-UNLIMITED TASKS

In tasks that are not time-limited by nature, the learning objective is to optimize the expectation of the return $G_0^\gamma$ from Equation 1. While the agent has to maximize its expected return over an indefinite (possibly infinite) period, it is desirable to still use time limits in order to frequently reset the environment and increase the diversity of the agent's experiences. A common mistake, however, is to then consider the terminations due to such time limits as environmental ones. This is equivalent to optimizing for returns $G_{0:T}^\gamma$ (Equation 2), not accounting for the possible future rewards that could have been experienced if no time limits were used.

In the case of value-based algorithms, we propose to continue bootstrapping at states where termination is due to the time limit. The state-value function of a policy (from Equation 3) can be rewritten in terms of the time-limited return $G_{t:T}^\gamma$ and the bootstrapped value from the last state $v_\pi(S_T)$:

$$v_\pi(s) \doteq \mathbb{E}_\pi \left[ G_{t:T}^\gamma + \boldsymbol{\gamma^{T-t} v_\pi(S_T)} | S_t = s \right] \tag{5}$$

By denoting an estimate of the state-value function by $\hat{v}_\pi$, the temporal-difference update rule after transitioning from a state $s$ to a state $s'$ and receiving a reward $r$ as a result of an action $a$ is given by either of the following expressions conditioned on the time step $t$:

$$\begin{aligned} \hat{v}_\pi(s) &\leftarrow (1-\alpha)\hat{v}_\pi(s) + \alpha\left(r + \gamma\hat{v}_\pi(s')\right) && \text{for } t < T-1 \\ \hat{v}_\pi(s) &\leftarrow (1-\alpha)\hat{v}_\pi(s) + \alpha\left(r + \boldsymbol{\gamma\hat{v}_\pi(s')}\right) && \text{for } t = T-1 \end{aligned} \tag{6}$$

The proposed partial-episode bootstrap is indicated in bold and green. An agent without this modification would update $\hat{v}_\pi(s)$ with or without bootstrapping from the estimated value of $s'$ depending on whether there is some remaining time or not. Similarly to Equation 4, the conflicting updates for estimating the value of the same state leads to an approximate average of these updates.

While this section is related to the previous one, it is somewhat orthogonal. In the previous section, one of the issues was bootstrapping values from states that were out-of-reach, letting the agent falsely believe that more rewards were available after. On the opposite, the problem presented here is when systematic bootstrapping is not performed from states at the time limit and thus, forgetting that more rewards would actually be available thereafter.

### 3.1 THE TWO-GOALS GRIDWORLD PROBLEM

We revisit the gridworld environment from Section 2.2. While previously the agent's task was to learn an optimal policy for a given time limit, we now consider how an agent can learn a good policy for an indefinite period from partial-episode experiences. We use the same setup as in Section 2.2. Again, we use a tabular Q-learning, but instead of considering terminations due to time limits as environmental ones, we continue bootstrapping from the non-terminal states that are reached at the time limits. This modification allows our agent to learn the time-unlimited optimal policy of always going for the most rewarding goal (see Figure 1c). On the other hand, while the standard agent that is not performing the final bootstrapping (see Figure 1a) had values from out-of-reach cells leaking into its learned value function, these bootstraps did not occur in sufficient proportion with respect to the terminations due to time limits in order to let the agent learn the time-unlimited optimal policy.

For the next experiments, we again use PPO but with two key modifications. We modify the Gym's TimeLimit wrapper to *not* include the remaining time (as needed for Section 2), but instead to include a flag to differentiate the terminations due to the time limit from the environmental ones. We also modify the PPO's implementation to enable continuing to bootstrap when terminations are due to time limits only. This involves modifying the implementation of the generalized advantage estimation (GAE) (Schulman et al., 2016). While GAEs use an exponentially-weighted average of $n$-step value estimations for bootstrapping that are more complex than the one-step lookahead bootstrapping explained in Equation 6, continuing to bootstrap from the last non-terminal states (i.e. at the end of the partial-episodes) is the only modification required for the proposed approach.

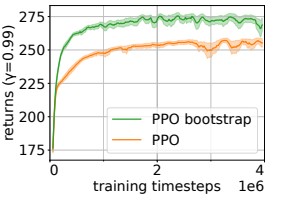 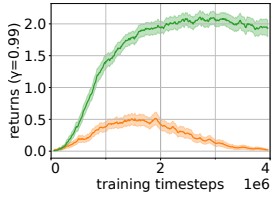

(a) Hopper-v1 ($T = 200$)   (b) InfiniteCubePusher ($T = 50$)

Figure 5: Performance evaluations of PPO with and without partial-episode bootstrapping. (a) On Hopper-v1 with $T = 200$ during the training and $T = 10^6$ during the evaluations. (b) On Infinite-CubePusher with $T = 50$ during the training and $T = 10^3$ during the evaluations. The standard PPO agent degrades drastically after some time.

### 3.2 HOPPER

Here, we consider the Hopper-v1 environment from Section 2.4, but instead aim to learn a policy that maximizes the agent's expected return over a time-unlimited horizon. We do not revisit the Reacher-v1 and the InvertedPendulum-v1 environments as their extensions to time-unlimited domains is not of particular value—that is, staying at the target position long after the target is reached (Reacher-v1) or maintaining the pendulum's vertical pose long after it is balanced (InvertedPendulum-v1). The aim here is to show that by continuing to bootstrap from episode terminations that are due to time limits only, we are able to learn good policies for time-unlimited domains. Figure 5a demonstrates performance evaluations of the standard PPO against one with the proposed partial-episode boot-strapping modification. The agents are trained on time-limited episodes of maximum $T = 200$ time steps, and are evaluated in the same environment, but with $T = 10^6$ time steps. We show that the proposed bootstrapping method significantly outperforms the standard PPO. During the evaluations, the standard PPO agent managed to reach a maximum of 7238 time steps on only one of the 40 training seeds, while our agent managed to reach the evaluation time limit of $T = 10^6$ on 7 occasions. This is quite impressive as this time limit corresponds to more than 2 hours of rendered hopping.

### 3.3 THE INFINITE CUBE PUSHER TASK

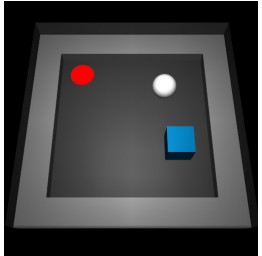

The proposed bootstrapping at time-limit terminations was shown to enable our agent to effectively learn a good long-term policy on Hopper-v1. However, it could be argued that since the Hopper-v1 environment always starts in quite similar configurations, the resultant policy is overfitted and almost completely cyclic. To demonstrate the ability of our proposed agent in learning non-cyclic policies for time-unlimited domains, we create a novel MuJoCo environment consisting of a torque-controlled ball that has to be used to push a cube to specified target positions. Once the cube has touched the target, the agent is rewarded and the target is moved away from the cube to a new random position. Because the task lacks terminal states, it can continue indefinitely. The objects are surrounded by fixed bounding walls. The inner edge of the walls stops the cube but not the ball in order to let the agent move the cube even if it is in a corner. The movements of the ball are limited to the horizontal plane and to the area defined by the outer edge of the walls. The environment's state representation consists of the objects' coordinates and velocities, and the cube's rotation. The agent receives no rewards unless the cube reaches a target location, at which point the agent receives a reinforcement reward of 1.

Due to the absence of reward shaping, reinforcement learning agents are prone to being stuck, unable to learn to solve problems. Thus it is often useful to introduce a time limit during training in order to facilitate learning. We use a training time limit of 50, only sufficient to push the cube to one target location in most cases. The evaluations, however, consisted of $10^3$ steps, long enough to allow successfully reaching several targets. Figure 5b shows the performance comparison of the standard PPO against one with the proposed modification. An entropy coefficient of 0.03 is used to encourage exploration and higher the chance of reaching a target and experiencing a reinforcement reward. We found this value to yield best performance for both agents among those from the set $\{0, 0.01, 0.02, 0.03, 0.04, 0.05\}$. While the performance of the standard PPO degrades significantly

after some time, it is clear that bootstrapping at the time limit helps our agent to perform significantly better. The maximum number of targets reached by our agent in a single episode of evaluation ($T = 10^3$) was 36 against 21 for the standard PPO.

## 4 DISCUSSION

We showed in Section 2 that time-awareness is required for correct credit assignment in domains where the agent has to optimize its performance over a time-limited horizon. However, even without the knowledge of the remaining time, reinforcement learning agents still often manage to perform relatively well. This could be due to several reasons including: (1) If the time limit is sufficiently long that terminations due to time limits are hardly ever experienced—for instance, in the Arcade Learning Environment (ALE) (Bellemare et al., 2013; Machado et al., 2017) domains where $T = 5$ minutes of interaction time is perhaps sufficiently long in most cases. (2) If there are clues in the environment that are correlated with the remaining time—for example, time-dependant attributes such as the forward distance. (3) If the relationship among the state-dependent action-advantages remains preserved. (4) If state-aliasing due to unawareness of the remaining time does not occur because it is not likely to observe the same states at different remaining times. (5) If the discount factor is sufficiently small to reduce the impact of the confusion. Furthermore, many methods exist to handle POMDPs (Lovejoy, 1991). In deep learning (LeCun et al., 2015; Schmidhuber, 2015), it is highly common to use a stack of previous observations or recurrent neural networks (RNNs) (Goodfellow et al., 2016) to address scenarios with partial observations (Wierstra et al., 2009). These solutions may to an extent help when the remaining time is not included as part of the agent's input. However, they are much more complex architectures and are only next-best solutions, while including a notion of the remaining time is quite simple and allows better diagnosis of the learned policies. The proposed approach is quite generic and can potentially be applied to domains with varying time limits where the agent has to learn to generalize as the remaining time approaches zero. In real-world applications such as robotics the proposed approach could easily be adapted by using the real time instead of simulation time steps.

In order for the proposed partial-episode bootstrapping in Section 3 to work, as is the case for value-based methods in general, the agent needs to bootstrap from reliable estimated predictions. This is in general resolved by enabling sufficient exploration. However, when the interactions are limited in time, exploration of the full state-space may not be feasible from some fixed starting states. Thus, a good way to allow appropriate exploration in such domains is to sufficiently randomize the initial states. It is worth noting that the proposed partial-episode bootstrapping is quite generic in that it is not restricted to partial episodes caused only due to time limits. In fact, this approach is valid for any early termination causes. For instance, it is common in the curriculum learning literature to start from near the goal states (easier tasks), and gradually expand to further states (more difficult tasks) (Florensa et al., 2017). In this case, it can be helpful to stitch the learned values by terminating the episodes and bootstrapping as soon as the agent enters a state that is already well known.

Since the proposed methods were shown to enable to better optimize for the time-limited and time-unlimited domains, we believe that they have the potential to improve the performance and stability of a large number of existing reinforcement learning algorithms. We also propose that, since reinforcement learning agents are in fact optimizing for the expected returns, and not the undiscounted sum of rewards, it is more appropriate to consider this measure for performance evaluation.

## 5 CONCLUSION

We considered the problem of learning optimal policies in time-limited and time-unlimited domains using time-limited interactions. We showed that when learning policies for time-limited tasks, it is important to include a notion of the remaining time as part of the agent's input. Not doing so can cause state-aliasing which in turn can result in suboptimal policies, instability, and slower convergence. We then showed that, when learning policies that are optimal for time-unlimited tasks, it is more appropriate to continue bootstrapping at the end of the partial episodes when termination is due to time limits, or any other early termination causes other than environmental ones. In both cases, we illustrated that our proposed methods can significantly improve the performance of PPO and allow to optimize more directly, and accurately, for either of the optimality models.

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

# SUPPLEMENTARY MATERIAL:
# TIME LIMITS IN REINFORCEMENT LEARNING

## A  ALL RESULTS FOR TIME-AWARE PPO

### A.1  QUEUE OF CARS

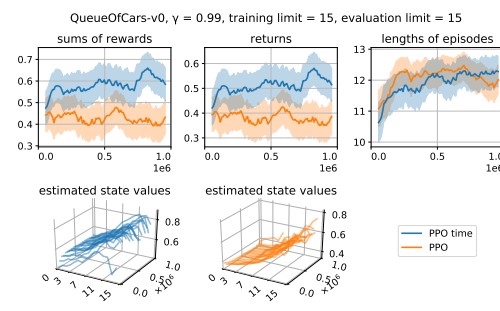
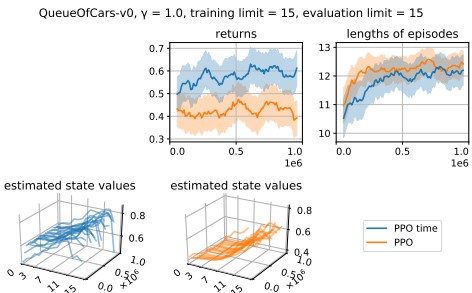

### A.2  INVERTEDPENDULUM-v1

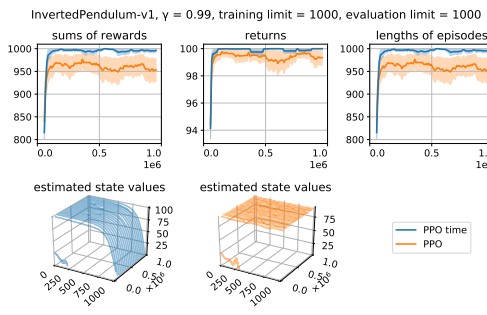
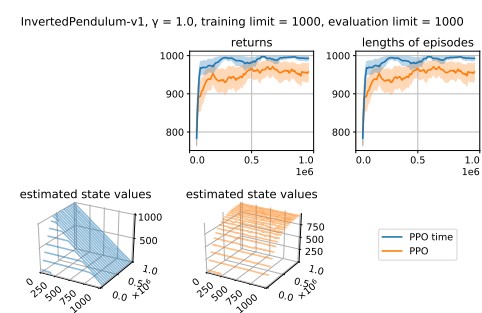

### A.3  REACHER-v1

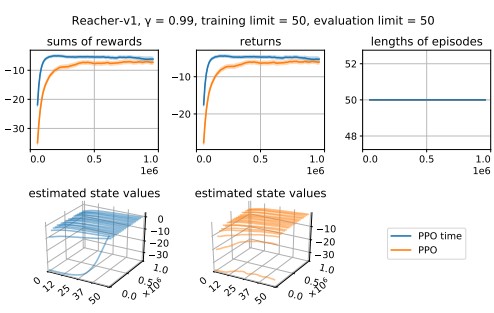
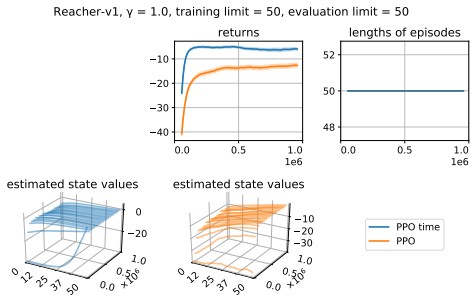

## A.4 HOPPER-V1

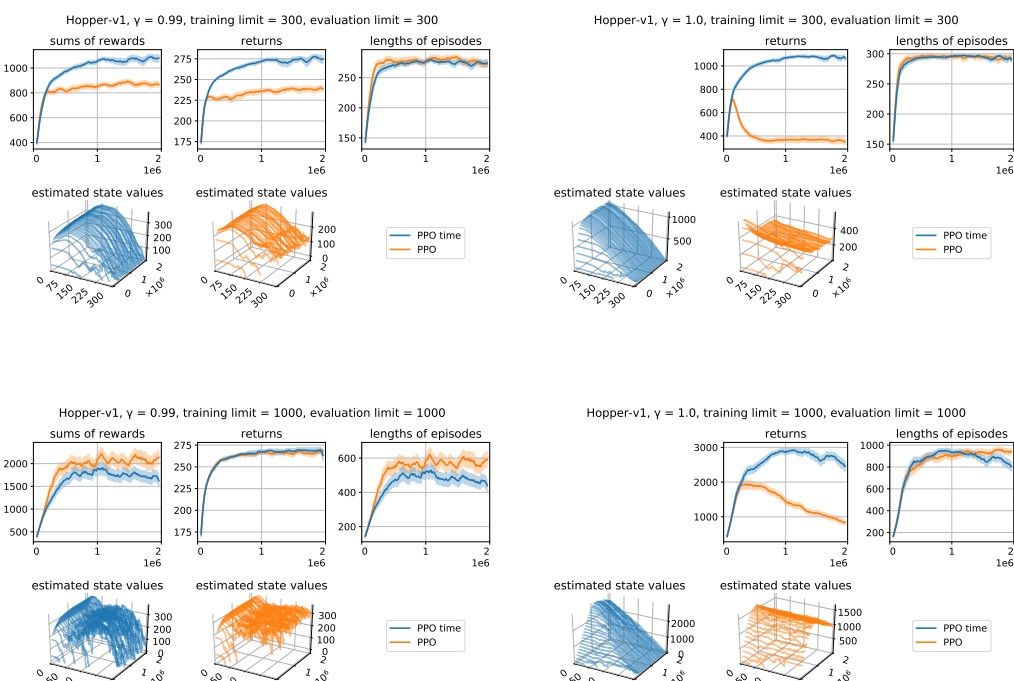

# B  ALL RESULTS FOR PPO WITH PARTIAL-EPISODE BOOTSTRAPPING

## B.1  HOPPER-V1[2]

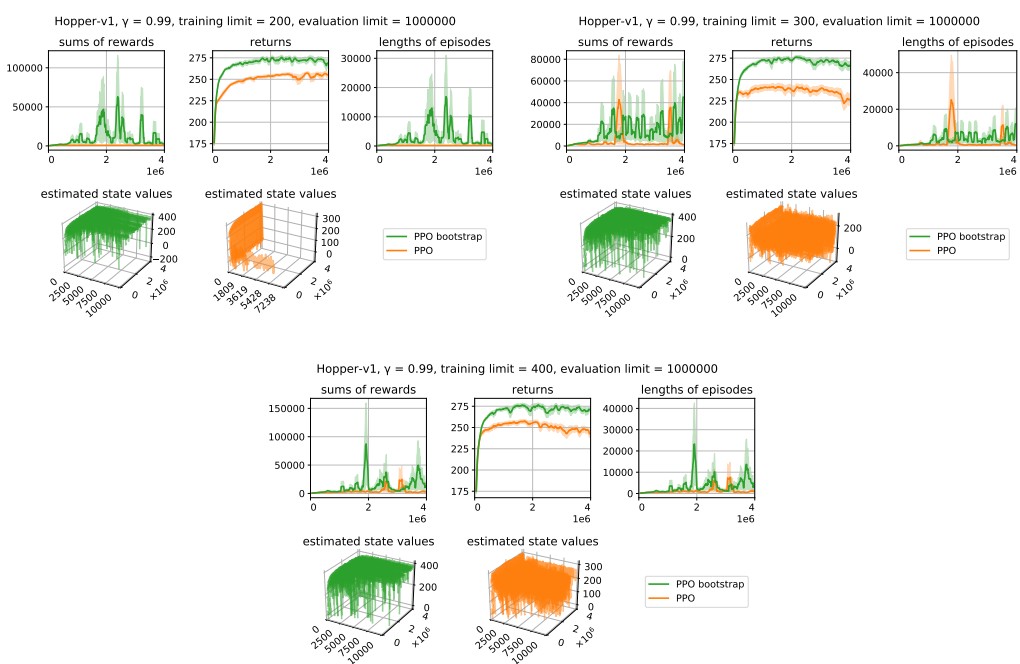

## B.2  INFINITE CUBE PUSHER

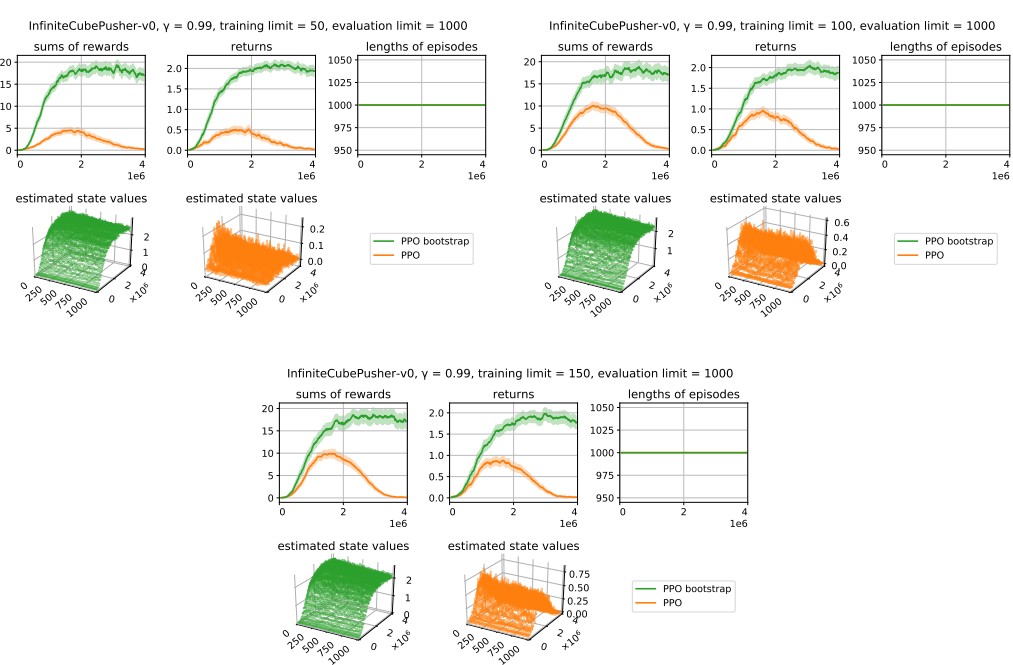

---

[2]For the estimated state-value graphs, the episode time steps are only shown up to 10,000.

