# OpenReview forum: "Time Limits in Reinforcement Learning"
_ICLR.cc/2018/Conference — Reject_

### Official Review · AnonReviewer1 · 2017-11-27
**Experimenting with time and terminations in reinforcement learning**

**Rating:** 5
**Confidence:** 4

**Review:**

Summary: This paper explores how to handle two practical issues in reinforcement learning. The first is including time remaining in the state, for domains where episodes are cut-off before a terminal state is reached in the usual way. The second idea is to allow bootstrapping at episode boundaries, but cutting off episodes to facilitate exploration. The ideas are illustrated through several well-worked micro-world experiments.

Overall the paper is well written and polished. They slowly worked through a simple set of ideas trying to convey a better understanding to the reader, with a focus on performance of RL in practice.

My main issue with the paper is that these two topics are actually not new and are well covered by the existing RL formalisms. That is not to say that an empirical exploration of the practical implications is not of value, but that the paper would be much stronger if it was better positioned in the literature that exists.

The first idea of the paper is to include time-remaining in the state. This is of course always possible in the MDP formalism. If it was not done, as in your examples, the state would not be Markov and thus it would not be an MDP at all. In addition, the technical term for this is finite horizon MDPs (in many cases the horizon is taken to be a constant, H). It is not surprising that algorithms that take this into account do better, as your examples and experiments illustrate. The paper should make this connection to the literature more clear and discuss what is missing in our existing understanding of this case, to motivate your work. See Dynamic Programming and Optimal Control and references too it.

The second idea is that episodes may terminate due to time out, but we should include the discounted value of the time-out termination state in the return. I could not tell from the text but I assume, the next transition to the start state is fully discounted to zero, otherwise the value function would link the values of S_T and the next state, which I assume you do not want. The impact of this choice is S_T is no longer a termination state, and there is a direct fully discounted transition to the start states. This is in my view is how implementations of episodic tasks with a timeout should be done and is implemented this way is classic RL frameworks (e.g., RL glue). If we treat the value of S_T as zero or consider gamma on the transition into the time-out state as zero, then in cost to goal problems the agent will learn that these states are good and will seek them out leading to suboptimal behavior. The literature might not be totally clear about this, but it is very well discussed in a recent ICML paper: White 2017 [1]

Another way to pose and think about this problem is using the off-policy learning setting---perhaps best described in the Horde paper [2]. In this setting the behavior policy can have terminations and episodes in the classic sense (perhaps due to time outs). However, the agent's continuation function (gamma : S -> [0,1]) can specify weightings on states representing complex terminations (or not), completely independent of the behavior policy or actual state transition dynamics of the underlying MDP. To clearly establish your contributions, the authors must do a better job of relating their work to [1] and [2].

[1] White. Unifying task specification in reinforcement learning. Martha White. International Conference on Machine Learning (ICML), 2017.

[2] Sutton, R. S., Modayil, J., Delp, M., Degris, T., Pilarski, P. M., White, A., & Precup, D. (2011). Horde: A scalable real-time architecture for learning knowledge from unsupervised sensorimotor interaction. In The 10th International Conference on Autonomous Agents and Multiagent Systems: 2, 761--768.

Small comments that did not impact paper scoring:
1) eq 1 we usually don't use the superscript \gamma
2) eq2, usually we talk about truncated n-step returns include the value of the last state to correct the return. You should mention this
3) Last paragraph of page 2 should not be in the intro
4) in section 2.2 why is the behavior policy random instead of epsilon greedy?
5) It would be useful to discuss the average reward setting and how it relates to your work.
6) Fig 5. What does good performance look like in this domain. I have no reference point to understand these graphs
7) page 9, second par outlines alternative approaches but they are not presented as such. Confusing

---

> ### Author Response · Authors · 2018-01-05
> **Clarifications for AnonReviewer1**
>
> We are grateful for the valuable feedback. Below is our response:
>
> We agree with the fact that the paper should be better positioned in the existing literature. Considering the first part of the paper on time-awareness, literature in dynamic programming and optimal control [3] generally suggests either a model-based backward induction method or to learn value functions for each time step. We feel that considering time as part of the MDP states and including it in the agent observations in the RL setting has been largely overlooked. This oversight has affected the design of the current benchmarks and we show that accounting for it can significantly improve the performance. We have also shown the specific effects time-based state-aliasing has on the learned value functions and policies of the agents and how discounting helps to somewhat mitigate this issue.
>
> The provided reference [1] introduces a way to consider episodic tasks as a continuing one with a variable discount factor to account for terminations in episodic tasks. The paper then uses the framework to introduce soft-terminations, where the terminal state of an episode retains part of the value of the starting state of the next episode by using a non-zero discount factor at the terminal state. Such an approach is different to ours in several ways:
> - Their framework is mainly valuable when episodes can be identified inside of a continuing task e.g. in the Taxi task, pick up and drop off passengers, similar to pushing cubes to targets in our InfiniteCubePusher task, however we show that our proposed partial-episode bootstrapping also makes sense in tasks that do not have an underlying episodic structure such as for Hopper.
> - Their framework do not permit environmental reset for soft-terminations, as the state after soft-termination may not be related. Our approach does not update the value of the last state of partial episodes, thus the transition between this one and the starting state of  the new episode does not exist in the view of the agent.
> - Soft-terminations use a different discount factor to the rest of the updates. We bootstrap with the same discount factor, showing in the Two-Goal Gridworld task that a correct indefinite-time value function can be learned from short episodes, just as it would be from indefinite time episodes.
> An advantage of the method proposed in the reference [1] and illustrated with the Taxi domain is that it encourages termination in a state that is good to start the next episode. If we take our InfiniteCubePusher task, this means, e.g. to not lose control of the cube once moved to the target by pushing it too hard and waste time for the next episode. With our approach, we too achieve this objective, as can be seen in a video linked in the paper (https://www.youtube.com/watch?v=ckgVLgFi-sc).
>
> As proposed in the Discussion section, partial-episode bootstrapping could be extended to some more sophisticated early-termination reasons, e.g. when an already well-known state is encountered. Furthermore, the proposed approach is compatible with on- or off-policy algorithms as the termination is not decided by the agent.
>
> Below is our response to some additional comments:
>
> 4) Since we are using an off-policy method (Q-learning), having a fully exploratory behavior policy does not prevent the agent from learning the optimal policy.
>
> 6) The video linked in the paper (https://www.youtube.com/watch?v=ckgVLgFi-sc) shows a learned good policy that manages to push the cube to several targets. For further information on the average number of targets reached during evaluation, you may refer to Section B.2. Since the rewards are 1 for reaching a target and 0 otherwise, thus the sums of rewards correspond to the average number of targets reached (approximately 17 during 1000 time steps for the proposed agent).
>
> [3] Bertsekas, Dimitri P., Dynamic programming and optimal control. Belmont, MA: Athena scientific, 2017.

---

### Official Review · AnonReviewer2 · 2017-12-01
**already known**

**Rating:** 4
**Confidence:** 5

**Review:**

The majority of the paper is focused on the observation that (1) making policies that condition on the time step is important in finite horizon problems, and a much smaller component on that (2) if episodes are terminated early during learning (say to restart and promote exploration) that the values should be bootstrapped to reflect that there will be additional rewards received in the true infinite-horizon setting.

1 is true and is well known. This is typically described as finite horizon MDP planning and learning and the optimal policy is well known to be nonstationary and depend on the number of remaining time steps. There are a number of papers focusing on this for both planning and learning though these are not cited in the current draft.

I don’t immediately know of work that suggests bootstrapping if an episode is terminated early artificially during training but it seems a very reasonable and straightforward thing to do.

---

> ### Author Response · Authors · 2018-01-05
> **Clarifications for AnonReviewer2**
>
> We are grateful for the valuable feedback. Below is our response:
>
> We agree with the fact that the paper should be better positioned in the existing literature. Considering the first part of the paper on time-awareness, literature in dynamic programming and optimal control [1] generally suggests either a model-based backward induction method or to learn value functions for each time step. We feel that considering time as part of the MDP states and including it in the agent observations in the RL setting has been largely overlooked. This oversight has affected the design of the current benchmarks and we show that accounting for it can significantly improve the performance. We have also shown the specific effects time-based state-aliasing has on the learned value functions and policies of the agents and how discounting helps to somewhat mitigate this issue.
>
> [1] Bertsekas, Dimitri P., Dynamic programming and optimal control. Belmont, MA: Athena scientific, 2017.

---

### Official Review · AnonReviewer4 · 2017-12-01
**Well written but not enough substance**

**Rating:** 4
**Confidence:** 4

**Review:**

This paper considers the problem of Reinforcement Learning in time-limited domains. It begins by observing that in time-limited domains, an agent unaware of the remaining time can experience state-aliasing. To combat this problem, the authors suggest modifying the state representation of the policy to include an indicator of the amount of remaining time. The time-aware agent shows improved performance in a time-limited gridworld and several control domains. Next, the authors consider the problem of learning a time-unlimited policy from time-limited episodes. They show that by bootstrapping from the final state of the time-limited domain, they are able to learn better policies for the time-unlimited case.

Pros:
The paper is well-written and clear, if a bit verbose.
The paper has extensive experiments in a variety of domains.

Cons:
In my opinion, the substance of the contribution is not enough to warrant a full paper and the problem of time-limited learning is not well motivated:

1) It's not clear how frequently RL agents will encounter time-limited domains of interest. Currently most domains are terminated by failure/success conditions rather than time. The author's choice of tasks seem somewhat artificial in that they impose time limits on otherwise unlimited domains in order to demonstrate experimental improvement. Is there good reason to think RL agents will need to contend with time-limited domains in the future?

2) The inclusion of remaining-time as a part of the agent's observations and resulting improvement in time-limited domains is somewhat obvious. It's well accepted that in any partially observed domain, inclusion of the latent variable(s) as a part of the agent's observation will result in a fully observed domain, less state-aliasing, more accurate value estimates, and better performance. The author's inclusion of the latent time variable as a part of the agent's observations reconfirms this well-known fact, but doesn't tell us anything new.

3) I have the same questions about Partial Episode bootstrapping: Is there a task in which we find our RL agents learning in time-limited settings and then evaluated in unlimited ones? The experiments in this direction again feel somewhat contrived by imposing time limits and then removing them. The proposed solution of bootstrapping from the value of the terminal state v(S_T) clearly works, and I suspect that any RL-practitioner faced with training time-limited policies that are evaluated in time-unlimited settings might come up with the same solution. While the experiments are well done, I don't think the substance of the algorithmic improvement is enough.

I think this paper would improve by demonstrating how time-aware policies can help in domains of interest (which are usually not time-limited). I could imagine a line of experiments that investigate the idea of selectively stopping episodes when the agent is no longer experiencing useful transitions, and then showing that the partial episode bootstrapping can save on overall sample complexity compared to an agent that must experience the entirety of every episode.

---

> ### Author Response · Authors · 2018-01-05
> **Clarifications for AnonReviewer4**
>
> We are grateful for the valuable feedback. Below is our response:
>
> 1) We disagree with the assertion that most domains are terminated by failure/success with no time limit. Many tasks can consist of maximizing a score within a time limit; e.g. robotic vacuum cleaning, taking an exam, or manufacturing where appropriate schedule is of essence. In OpenAI Gym, currently featuring the most popular set of benchmark environments, all the tasks are time-limited. In fact, many of the environments never terminate before the time limit, such as Reacher, Pusher, Striker, Thrower, Swimmer, HumanoidStandup, and Pendulum. Furthermore, even in environments that can terminate by failure, such as InvertedPendulum or Hopper, we show that time-awareness is valuable.
>
> 2) Indeed time-awareness seems somewhat obvious and solves the state-aliasing due to time, but since it has been largely overlooked in the RL literature, we believe it was necessary to produce a comprehensive account. Moreover, the role of state-aliasing merely as a consequence of not observing a notion of the remaining time has not been explicitly discussed previously. We elaborate on this scenario and the specific issues it causes. We stand to believe that the RL community could indeed benefit from such clarification.
>
> 3) Early termination of episodes can be helpful, or even necessary, to an RL algorithm; specifically, if an agent gets stuck in some parts of the state space because of traps in the environment or poor exploration. We have encountered this issue for the InfiniteCubePusher task with PPO. The agent would very commonly learn to permanently push against the wall if the environment was never reset. However, if early terminations are used, we show that it is possible to learn a good long-term policy by continuing to bootstrap at early termination. This approach, however intuitive, is not discussed explicitly in any literature to the best of our knowledge. In fact, as we show in the paper, available implementations of the state-of-the-art algorithms are not taking this into consideration. Therefore, we believe such an account to bring clarity to the field is valuable to the community.
>
> 4) Indeed, early-termination based on known states instead of time limits is very interesting and is proposed in the discussion section. However, to make a coherent paper solving a specific and very recurrent problem in the RL literature, we decided to focus on time limits.

---

### Public Comment · ~Clayton_Thorrez1 · 2017-12-21
**Reproducibility report on this paper**

As part of the ICLR Reproducibility Challenge, we have made an attempt to reproduce some of the experiments found in this paper.

We started from the OpenAI baseline inlementation of PPO as did the authors of this paper and we modified the environment wrappers to allow for the time remaining in an episode to be used as a feature in the observation space. We tested this time aware agent on Hopper, InvertedPendulum and Reacher.

The authors of this paper made strong efforts to make their work reproducible. They starte their work building off of a publicly available code base and several of their experiments were on common Mujoco robot environments. All hyperparameters, the training time, the random seeds and the specifics of smoothing for graphs were reported.

We did our best to make the same changes to the OpenAI enrivonment wrapper to allow for time remaining to be a feature of the opservation space and thus used as a feature in the neural network for prediction.

We successfully ran experiments with the time aware agent on the Hopper, Reacher and InvertedPendulum environments.

On Hopper with gamma = 0.99, our results closely matched those reported in the paper.  There was a clear benefit to using time as a feature. However for undiscounte rewards with gamma set to 1, our result showed the same advantage for time awareness as with gamma = 0.99. The original authors reported a sharp decline in rewards for the time unaware agent.

On InvertedPendulum, within bound of uncertainty, our results seem to confirm the results reported in this paper for both the time aware and time unaware agents with gamma set to both 0.99 and to 1.

On Reacher, the authors reported an advantage for time awareness at early states in training for epilon = 0.99 and a very strong advantage for time awareness at all stages when gamma = 1. Our results did not reflect this. Our experiment indicated roughly equal performance for time aware and time unaware agents for both gamma = 0.99 and gamma = 1.

In conclusion, there are clear signs of advantages of time awareness on some environments. However, our implementation failed to realizefully those advantages on all environments.

Our full report may be read here: https://drive.google.com/file/d/1wiVVj_zSg4t-w8x6LHBYhHkxF5Est2SX/view?usp=sharing

Edited: epsilon -> gamma

---

> ### Comment · AnonReviewer1 · 2017-12-21
> **gamma not epsilon**
>
> I think you mean gamma=0.99 above

---

> > ### Public Comment · ~Clayton_Thorrez1 · 2017-12-21
> > **You are correct**
> >
> > You are right, thank you for the correction.

---

> ### Author Response · Authors · 2018-01-05
> **Clarifications for the reproducibility report**
>
> Thank you kindly for your work reproducing the first part of the paper. We believe this is a very important initiative for the research in the field and we indeed did our best to make the paper reproducible.
>
> As you mention in the report, you used Roboschool while our results were obtained using the more popular Gym MuJoCo environments. The differences between the environments could certainly explain the contrast in our results.
>
> However, in order to illustrate that our figures can be easily reproduced, we created a very simple notebook that shows similar results as the ones you tried to replicate. For simplicity, the notebook is using the average of the last 100 rewards collected during training, which is different from the way we evaluated the performance in the paper (one complete episode every five training cycles) and averaged it (sliding window and 40 seeds). The notebook can be found here:
> https://gist.github.com/anonymous/dfd23c90a3bb69b650d76f690d6cd501

---

### Decision · Program_Chairs · 2018-01-29
**ICLR 2018 Conference Acceptance Decision**

**Decision:**

Reject

**Comment:**

The reviewers agree that this paper suffers from a lack of novelty and does not make sufficient contributions to warrant acceptance.